# Kernel-based Unsupervised Embedding Alignment for Enhanced Visual Representation in Vision-language Models

**Shizhan Gong** [1]   **Yankai Jiang** [2]   **Qi Dou** [1]   **Farzan Farnia** [1]

## Abstract

Vision-language models, such as CLIP, have achieved significant success in aligning visual and textual representations, becoming essential components of many multi-modal large language models (MLLMs) like LLaVA and Open-Flamingo. However, numerous studies have identified CLIP's limited fine-grained perception as a critical drawback, leading to substantial failures in downstream MLLMs. In contrast, vision-centric foundation models like DINOv2 demonstrate remarkable capabilities in capturing fine details from images. In this work, we propose a novel kernel-based method to align CLIP's visual representation with that of DINOv2, ensuring that the resulting embeddings maintain compatibility with text embeddings while enhancing perceptual capabilities. Our alignment objective is designed for efficient stochastic optimization. Following this image-only alignment fine-tuning, the visual encoder retains compatibility with the frozen text encoder and exhibits significant improvements in zero-shot object recognition, fine-grained spatial reasoning, and localization. By integrating the aligned visual encoder, downstream MLLMs also demonstrate enhanced performance. The code and models are available at https://github.com/peterant330/KUEA.

## 1. Introduction

Vision-language Models (VLMs) have made significant strides and transformed the field of computer vision. A notable example is CLIP (Radford et al., 2021) and its variants (Zhai et al., 2023; Sun et al., 2023), which are trained on extensive datasets of paired text and images to link images with their corresponding textual descriptions. These models demonstrate exceptional generalizability and zero-shot performance on various downstream tasks, including classification (Saha et al., 2024), segmentation (Yu et al., 2023), and object detection (Vidit et al., 2023). Beyond functioning as a standalone tool, CLIP's vision encoder has been incorporated into several Multi-modal Large Language Models (MLLMs), such as LLaVA (Liu et al., 2024), Open-Flamingo (Awadalla et al., 2023), BLIP-2 (Li et al., 2023a), and Qwen-VL (Bai et al., 2023), serving as an integral component for visual feature extraction.

However, because CLIP is trained with global supervision from image captions, it struggles to learn finer pixel-level details such as color (Thrush et al., 2022) and spatial relationship (Liu et al., 2023; Kamath et al., 2023). This limitation affects CLIP's performance on vision-focused tasks and can also impair the fine-grained perception capabilities of downstream MLLMs. Numerous studies have highlighted these issues (Yuksekgonul et al., 2023; Guo et al., 2024). For instance, Jiang et al. (2023) examined feature visualization from deeper layers and found that it emphasizes global image properties while neglecting intricate details. Similarly, Tong et al. (2024) reported that current MLLMs struggle with simple visual pattern questions, such as counting, color identification, and viewpoint recognition.

Vision-only self-supervised learning protocols, such as DINO (Caron et al., 2021) and MAE (He et al., 2022), produce vision-centric representations that are highly effective for visual grounding. Researchers are exploring these models to address the inherent limitations of CLIP. Some studies have combined multiple models as vision encoders for MLLMs (Jiang et al., 2023; Tong et al., 2024; Shi et al., 2024; Shen et al., 2024). However, this approach introduces considerable computational overhead. Other methods focus on using vision-only self-supervised learning to fine-tune the CLIP encoder (Wu et al., 2023; Covert et al., 2025), which can enhance its localization abilities. Nevertheless, these fine-tuning techniques risk disrupting the alignment between image and text representations, potentially under-

---

[1]Department of Computer Science and Engineering, The Chinese University of Hong Kong, Hong Kong SAR, China. [2]Shanghai Artificial Intelligence Laboratory, Shanghai, China. Correspondence to: Shizhan Gong <szgong22@cse.cuhk.edu.hk>, Yankai Jiang <jiangyankai@pjlab.org.cn>, Qi Dou <qidou@cuhk.edu.hk>, Farzan Farnia <farnia@cse.cuhk.edu.hk>.

*Proceedings of the 42nd International Conference on Machine Learning*, Vancouver, Canada. PMLR 267, 2025. Copyright 2025 by the author(s).

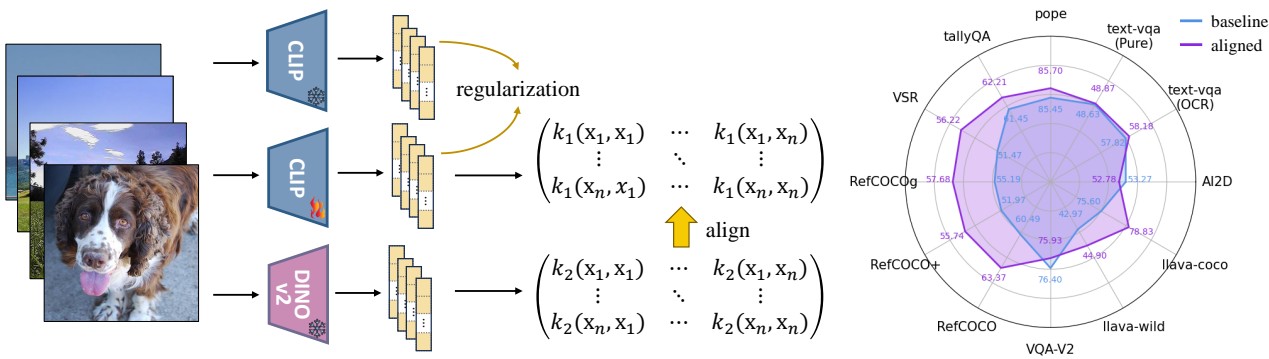

*Figure 1.* **Main claim of this work.** We propose a kernel-based alignment framework, which is able to enhance the visual representation of CLIP via image-only fine-tuning. Moreover, the improvement can be transferred to downstream multi-modal large language models.

mining CLIP's zero-shot performance. Additionally, there are efforts to integrate region-based task loss (Zhong et al., 2022; Wan et al., 2024) or distillation loss from vision models (Salehi et al., 2023) during CLIP's training, which requires re-training the models and incurs high computational costs. Most critically, all these methods produce visual embeddings that differ significantly from the original CLIP embeddings, leading to incompatibility with downstream models trained on the original embeddings. Consequently, all downstream models (e.g., MLLMs) would need to be re-tuned from scratch to align visual-text embeddings. The cost makes these solutions impractical in real applications.

In this paper, we present an innovative approach to address the limitations of CLIP embeddings by aligning them with embeddings from vision-centric models. Our method involves fine-tuning CLIP's vision branch to align with the representations of target models, such as DINOv2 (Oquab et al., 2024), while maintaining compatibility with CLIP's text embeddings, which is totally frozen and untouched (see Fig. 1). Given the significant differences between the feature spaces of target models and CLIP, a direct alignment of the visual representations could disrupt the alignment with text representations. Instead, we propose to align the embeddings in the kernel space, which preserves the integrity of the original feature space while allowing for flexible adjustments of similarities among samples based on their visual details. As a result, this alignment could enhance the vision encoder's ability to recognize fine-grained visual patterns. In addition, we design an optimization objective that can be handled by stochastic optimization, enabling our framework to scale effectively to real-world datasets with minimal computational hardware requirements.

We examine the enhancements achieved through the proposed alignment in both vision-centric tasks and visual question answering (VQA). Experiments conducted on multiple CLIP-benchmark (LAION-AI, 2022) and probing bench (Covert et al., 2025) reveal that the CLIP after alignment demonstrates improved accuracy in zero-shot classi-

fication and dense prediction tasks, without requiring fine-tuning of the text encoder. Subsequently, we integrate the aligned CLIP vision encoder into two pre-trained MLLMs, LLaVA (Liu et al., 2024) and OpenFlamingo (Awadalla et al., 2023), evaluating their performance across several standard VQA benchmarks. This also results in significant enhancements over the original CLIP, even without fine-tuning the large language model (LLM) component. Our main contributions can be summarized as follows.

- We introduce a kernel-based alignment method that effectively aligns two sets of embeddings while preserving the integrity of the original representation space.

- We implement the alignment using CLIP and DINOv2. With minimal image-only fine-tuning, the aligned CLIP visual encoder shows substantial improvements while preserving compatibility with the text encoder and ensuring zero-shot generalizability.

- Evaluation across various vision-language benchmarks shows that the enhancements in visual representations can be effectively inherited to downstream MLLMs.

## 2. Related Work

**Kernel Methods in Representation Learning.** Kernel methods (Hofmann et al., 2008) are algorithms that use the kernel trick, which allows linear algorithms to work in high-dimensional feature spaces without explicit data transformation. It is widely used in deep representation learning. For example, He & Ozay (2022) utilized kernel-based metrics for evaluating the similarity between two embeddings. Several work (Allen-Zhu & Li, 2020; He & Ozay, 2022; Zhou et al., 2024) leveraged kernel functions for knowledge distillation. Dehdashtian et al. (2024) proposed to de-bias CLIP's image and text representations in reproducing kernel Hilbert spaces for better fairness. Kernel methods have also been applied to evaluating the fidelity (Bińkowski et al., 2018), diversity (Friedman & Dieng, 2022; Jalali et al.,

2023; Rezaei et al., 2024; Ospanov & Farnia, 2024; Jalali et al., 2024; Ospanov et al., 2024a;b), and novelty (Zhang et al., 2024a;b) of generative models. Our method can be interpreted as aligning visual representations of CLIP and other vision models in the kernel space, through kernel trick.

**Fine-tuning Vision-Language Models.** Fine-tuning of VLMs has been widely applied to improve the model performance on specific downstream dataset (Zhou et al., 2022; Zhang et al., 2022; Gao et al., 2024) or domain-specific applications (Cao et al., 2024; Gong et al., 2024). Beyond the accuracy gain, studies have also discovered that well-designed fine-tuning scheme can promote some desired properties of the model, such as adversarial robustness (Mao et al., 2022; Schlarmann et al., 2024), fairness (Shen et al., 2023), and visual interpretability (Gong et al., 2025). In this work, we leverage fine-tuning to enhance the general visual ability of the VLMs. Instead of focusing on single datasets, we show the enhancement exhibit good generalizability.

**Enhancement of CLIP Visual Representations.** Several studies have proposed methods to address the limitations of CLIP's visual embeddings and enhance its fine-grained capabilities. For instance, Salehi et al. (2023) trained the CLIP encoder using multi-task losses supervised by pseudo labels generated from other vision encoders. Similarly, Jiang et al. (2023) and Tong et al. (2024) complemented CLIP with DINOv2 as the vision encoder for downstream MLLMs. Shi et al. (2024) and Shen et al. (2024) further expanded this approach to incorporate a wider variety of vision encoders. Currently, there are limited explorations focusing on the fine-tuning phase. Covert et al. (2025) applied masked fine-tuning to CLIP's vision encoder to enhance its localization capabilities. However, their fine-tuned vision encoders became incompatible with the text encoder and downstream LLMs. The method most closely related to ours is DIVA (Wang et al., 2025), which refines CLIP representations using only images with the help of diffusion models. Comparing to DIVA, our method demands significantly less computation and achieves better zero-shot performance.

# 3. Method

## 3.1. Kernel Function and Kernel Matrix

Consider a function $k : \mathbb{R}^d \times \mathbb{R}^d \to \mathbb{R}$ that assigns a similarity score $k(\mathbf{x}, \mathbf{y}) \in \mathbb{R}$ to every pair of vectors $\mathbf{x}, \mathbf{y} \in \mathbb{R}^d$. The function $k$ qualifies as a kernel function if and only if, for any set of samples $\mathbf{x}_1, \cdots, \mathbf{x}_n \in \mathbb{R}^d$, the resulting kernel matrix $\mathbf{K} = [k(\mathbf{x}_i, \mathbf{x}_j)]_{1 \le i,j \le n}$ is positive semi-definite. This kernel matrix well captures the pairwise similarities among the samples. Common examples of kernel functions include Gaussian kernel, cosine kernel, and polynomial kernel. In this work, we mainly focus on the polynomial kernel

with coefficient $\gamma$, constant offset $c$, and degree $d$:

$$k_{\text{polynomial}(\gamma,c,d)}(\mathbf{x}, \mathbf{y}) := (\gamma \mathbf{x}^T \mathbf{y} + c)^d. \quad (1)$$

We call a kernel function normalized if $k(\mathbf{x}, \mathbf{x}) = 1$ for any vector $\mathbf{x} \in \mathbb{R}^d$. A kernel function can be normalized via:

$$\tilde{k}(\mathbf{x}, \mathbf{y}) = \frac{k(\mathbf{x}, \mathbf{y})}{\sqrt{k(\mathbf{x}, \mathbf{x})k(\mathbf{y}, \mathbf{y})}}. \quad (2)$$

Note that for each valid kernel function $k$, there exists a feature map $\phi : \mathbb{R}^d \to \mathbb{R}^s$ such that for every input vectors $\mathbf{x}, \mathbf{y}$, the following holds:

$$k(\mathbf{x}, \mathbf{y}) = \langle \phi(\mathbf{x}), \phi(\mathbf{y}) \rangle, \quad (3)$$

where $\langle \cdot, \cdot \rangle$ denotes the inner product in $\mathbb{R}^s$. $s$ is usually much greater than $d$, indicating that $\phi$ maps the input features into a higher-dimensional kernel space.

## 3.2. Kernel-based Embedding Alignment

CLIP consists of two components: an image encoder $f_\theta$ and a text encoder $f_\varphi$. These encoders transform each language-image pair $(T_i, I_i)$ into their respective representations $f_\theta(I_i), f_\varphi(T_i) \in \mathbb{R}^d$. The target model $g$ is a vision encoder (e.g., DINOv2) that generates the representation $g(I_i) \in \mathbb{R}^{d'}$ for the image $I_i$. While CLIP effectively achieves visual-language alignment, it struggles to capture fine-grained visual details, an area where the target model excels. As noted by Tong et al. (2024), samples that carry similar semantic meanings but differ in visual details exhibit high similarity in the CLIP feature space, while showing low similarity in the DINOv2 feature space. The value of kernel function is a measure of similarity between two samples. The kernel matrix of the target model reflects how samples are arranged in feature spaces based on their visual similarities, whereas the kernel matrix of CLIP only captures semantic similarities. Consequently, it is a natural idea to align the kernel matrix of CLIP with that of the target model, as it can encourage the arrangement of samples in the feature space to better represent their visual patterns, thus alleviating the limitations of CLIP (as illustrated in Fig. 1). Due to the large sample size, directly minimizing the distance between the two kernel matrices can be computationally infeasible. To address this, we propose the following optimization objective for alignment fine-tuning:

$$\min_\theta \mathop{\mathbb{E}}_{I_i, I_j \sim \mathcal{D}_{\text{train}}} [(k_1(f_\theta(I_i), f_\theta(I_j)) - k_2(g(I_i), g(I_j)))^2], \quad (4)$$

where $k_1$ and $k_2$ are the kernel functions for CLIP and the target model, which may take different forms or with different parameters to reflect the variation in the feature space. According to Hoeffding's inequality (Hoeffding, 1994), the empirical estimate of the objective function is an unbiased estimator, and the following proposition implies that the gradient w.r.t $\theta$ can also be computed through limited samples.

*Table 1.* **Accuracy evaluation on zero-shot image classification benchmarks of CLIP models w/wo alignment**. *Projection* means training a linear layer to map DINOv2 representations into the CLIP feature space. *Feature* means directing aligning the representation pairs subjecting to a linear transformation. *DIVA* refers to the method proposed by Wang et al. (2025). *Kernel* is our proposed method.

| | Alignment Strategies | ImageNet | CIFAR10 | CIFAR100 | CalTech | FER | OxfordPets | DTD | RESISC | EuroSAT | PCAM | ImageNet-S | ImageNet-O | Average Zero-shot |
|---|---|---|---|---|---|---|---|---|---|---|---|---|---|---|
| | | | | | | | Zero-shot Datasets | | | | | | | |
| ViT-B-16 | CLIP | 67.72 | 89.98 | 65.61 | 82.17 | 46.35 | 89.04 | 44.95 | 58.19 | 55.89 | 50.70 | 48.28 | 42.30 | 61.22 |
| | projection | **68.96** | **96.24** | **75.30** | 73.67 | 27.19 | 80.70 | 34.26 | 33.27 | 40.44 | 50.02 | **54.73** | 30.00 | 54.17 |
| | feature | 67.84 | 90.47 | 66.46 | 82.19 | 46.36 | 89.02 | 45.00 | **58.29** | 56.04 | 50.79 | 48.31 | 42.60 | 61.41 |
| | kernel | 67.84 | 91.13 | 67.38 | **82.20** | **46.82** | **89.23** | **45.43** | 57.73 | **56.85** | **54.02** | 48.18 | **43.50** | **62.04** |
| ViT-L-14 | CLIP | 74.90 | 95.20 | 71.08 | 83.30 | 50.00 | 93.21 | 55.21 | 63.35 | 62.65 | 52.00 | 59.59 | 32.25 | 65.26 |
| | projection | 70.42 | 95.10 | 74.14 | 80.31 | 24.51 | 80.02 | 37.02 | 37.03 | 32.48 | 50.02 | 56.04 | 28.05 | 54.07 |
| | feature | 75.16 | 95.77 | 75.13 | 83.58 | 49.62 | 93.35 | 55.73 | 63.13 | 63.11 | 51.34 | 58.08 | 34.20 | 65.73 |
| | DIVA | 74.84 | 95.10 | 70.96 | 84.01 | 49.47 | **93.57** | 55.05 | 63.48 | 62.93 | **53.79** | 59.53 | 32.05 | 65.45 |
| | kernel | **75.52** | **96.27** | **77.04** | **84.32** | **50.31** | 93.40 | **55.74** | **63.83** | **63.83** | 52.68 | **59.67** | **34.90** | **66.54** |
| ViT-L-14-336 | CLIP | 75.82 | 94.49 | 71.11 | 83.42 | 49.04 | 93.70 | 55.64 | 63.73 | 61.46 | **60.68** | 61.03 | 32.75 | 66.10 |
| | projection | 71.26 | 94.87 | **77.35** | 72.33 | 23.35 | 79.39 | 39.20 | 42.06 | 35.06 | 50.02 | 57.53 | 28.65 | 54.53 |
| | feature | 75.36 | 95.45 | 74.16 | 83.45 | **50.24** | 93.79 | 56.22 | 64.37 | 61.22 | 59.82 | 60.40 | 32.55 | 66.52 |
| | DIVA | 75.54 | 93.74 | 70.73 | **83.63** | 50.07 | 93.32 | 55.48 | 63.19 | 60.13 | 59.52 | 56.90 | 31.65 | 65.30 |
| | kernel | **76.30** | **95.95** | 74.92 | 83.53 | 50.21 | **93.79** | **56.33** | **64.43** | **61.93** | 59.65 | **61.31** | **35.60** | **67.13** |

**Proposition 3.1.** *Assume that both $k_1$ and $k_2$ take values in the range $[-1, 1]$. Let $(I_{m_1}, I_{m_2})$ for $1 \leq m \leq M$ represent $M$ pairs of images independently sampled from the data distribution. Assume $k_1(f_\theta(I_{m_1}), f_\theta(I_{m_2}))$ is $L$-Lipschitz w.r.t. $\theta$ for any sampled pairs. Define the sample-wise gradient as*

$$t(\theta; I_{m_1}, I_{m_2})$$
$$:= \nabla_\theta \Big( k_1(f_\theta(I_{m_1}), f_\theta(I_{m_2})) - k_2(g(I_{m_1}), g(I_{m_2})) \Big)^2,$$

*and the true expected gradient is defined as $\mathbb{E}[t(\theta)] := \nabla_\theta \mathbb{E}_{I_i, I_j \sim \mathcal{D}_{train}}\Big[ \big(k_1(f_\theta(I_i), f_\theta(I_j)) - k_2(g(I_i), g(I_j))\big)^2 \Big]$. Then, for every $0 < \epsilon < 8L$, we have*

$$P\left( \left\| \frac{1}{M} \sum_{m=1}^{M} t(\theta; I_{m_1}, I_{m_2}) - \mathbb{E}[t(\theta)] \right\|_2 \geq \epsilon \right)$$
$$\leq \exp\left( -\frac{M\epsilon^2}{512L^2} + \frac{1}{4} \right) \tag{5}$$

As a result, we can utilize stochastic optimization methods, such as mini-batch gradient descent, to solve the problem efficiently. In each iteration, we sample a batch of data pairs and minimize the difference in kernel functions specifically for those pairs. This approach enables the alignment framework to scale effectively to datasets of real-world size.

### 3.3. Regularization for Visual-language Alignment

To ensure the alignment with the text branch is preserved, we introduce a regularization to prevent the aligned features

from straying too far from their original direction. This term is represented by the $L_2$ distance between the visual representations before and after the alignment fine-tuning. The final optimization objective is expressed as follows:

$$\min_\theta \; w \cdot \mathbb{E}_{I_i, I_j \sim \mathcal{D}_{train}} \Big[ \Big( k_1\big(f_\theta(I_i), f_\theta(I_j)\big) \tag{6}$$
$$- k_2\big(g(I_i), g(I_j)\big) \Big)^2 \Big] + \mathbb{E}_{I_i \sim \mathcal{D}_{train}} \Big[ \big\| f_\theta(I_i) - f_{\theta_0}(I_i) \big\|_2^2 \Big],$$

where $\theta_0$ is the parameters of the pre-trained CLIP and is frozen during the alignment phase. $w$ is a coefficient balancing the two loss terms. The regularization term ensures that the alignment process does not cause the aligned embeddings to deviate significantly from the original embeddings. The following proposition illustrates how this regularization helps preserve the language-image alignment.

**Proposition 3.2.** *(Schlarmann et al., 2024) For every language-image pair $(T, I)$, if $\|f_\theta(I) - f_{\theta_0}(I)\|_2 \leq \lambda$ holds, then we will have*

$$\left| \cos\big(f_{\theta_0}(I), g(T)\big) - \cos\big(f_\theta(I), g(T)\big) \right|$$
$$\leq \frac{2\lambda}{\max\{\|f_\theta(I)\|_2, \|f_{\theta_0}(I)\|_2\}}$$

The proposition shows this regularization helps preserve the language-image alignment without the need of incorporating any text data during the fine-tuning phase.

*Table 2.* **Summary of zero-shot image-to-text and text-to-image retrieval performance on Flickr30K (Young et al., 2014) and COCO benchmark (Chen et al., 2015) datasets.** Our alignment will not sacrifice the image-text alignment property of the original CLIP.

| | Vision Encoder | Image-to-Text Retrieval | | | | | | Text-to-Image Retrieval | | | | | |
| | | Flickr30K | | | MSCOCO | | | Flickr30K | | | MSCOCO | | |
| | | R@1 | R@5 | R@10 | R@1 | R@5 | R@10 | R@1 | R@5 | R@10 | R@1 | R@5 | R@10 |
|---|---|---|---|---|---|---|---|---|---|---|---|---|---|
| ViT-B-16 | w/o align | 77.90 | **94.30** | 97.40 | **48.36** | 72.68 | 81.76 | 60.64 | 83.82 | 90.36 | 31.80 | 55.91 | 66.97 |
| | w/ align | **78.10** | 94.10 | **97.50** | 48.06 | **72.86** | **81.78** | **60.92** | **84.30** | **90.62** | **32.12** | **56.51** | **67.15** |
| ViT-L-14 | w/o align | 81.40 | 96.20 | 98.70 | 50.64 | 74.20 | 82.96 | 63.62 | 86.36 | 91.86 | 34.51 | 59.21 | 69.37 |
| | DIVA | 78.70 | 94.80 | 98.30 | 49.88 | 74.06 | 82.82 | 59.80 | 83.36 | 89.40 | 34.21 | 58.76 | 68.97 |
| | w/ align | **83.00** | **96.90** | **99.00** | **51.72** | **75.52** | **83.30** | **64.78** | **87.20** | **92.32** | **35.98** | **61.08** | **71.02** |
| ViT-L-14-336 | w/o align | 83.00 | 96.60 | 99.00 | 52.12 | 76.12 | 83.82 | 64.78 | 87.92 | 93.06 | 35.65 | 60.30 | 70.66 |
| | DIVA | 80.20 | 95.90 | 98.30 | 52.34 | 76.48 | 84.16 | 61.20 | 84.72 | 90.80 | 35.77 | 60.29 | 70.44 |
| | w/ align | **84.60** | **96.80** | **99.10** | **53.48** | **77.64** | **85.30** | **67.08** | **88.98** | **93.62** | **37.61** | **62.48** | **72.46** |

*Table 3.* **Accuracy evaluation on counting, spatial reasoning, and caption recognition tasks** of CLIP models w/wo alignment.

| | Vision Encoder | svhn | gtsrb | clevr distance | clevr counts |
|---|---|---|---|---|---|
| Zero-shot | ViT-B-16 | **31.31** | 43.34 | 22.37 | 21.21 |
| | +align | 27.40 | **44.35** | **22.40** | **21.30** |
| | ViT-L-14 | 57.02 | 50.55 | 20.21 | 19.43 |
| | +align | **59.63** | **52.53** | **23.39** | **21.11** |
| | ViT-L-14-336 | 56.03 | 52.41 | 18.93 | 20.05 |
| | +align | **57.65** | **53.43** | **20.91** | **20.90** |
| Linear Probe | ViT-B-16 | 45.02 | 57.13 | **31.19** | 23.68 |
| | +align | **49.27** | **58.30** | 30.92 | **24.02** |
| | ViT-L-14 | 65.20 | 72.94 | 22.97 | 41.25 |
| | +align | **69.39** | **74.51** | **30.82** | **49.67** |
| | ViT-L-14-336 | 61.49 | 70.17 | 28.43 | 53.07 |
| | +align | **70.02** | **71.77** | **34.65** | **55.32** |

# 4. Experiments

## 4.1. Implementation Details

**Models.** The above framework is actually flexible enough to align any two sets of embeddings. In this study, we primarily examine the alignment of visual representations from CLIP and vision-focused models. We experiment with several versions of CLIP, including ViT-B-16, ViT-L-14, and ViT-L-14-336, all pre-trained by OpenAI. For the vision-centric target model, we employ DINOv2 (Oquab et al., 2024) with ViT-L-14 as the backbone, after registration (Darcet et al., 2024). This setup allows for aligned embedding pairs to originate from models with different architectures or to correspond to images of varying resolutions, demonstrating the flexibility and generalizability of our proposed framework.

**Training data.** We utilize the training set of ImageNet-1K (Deng et al., 2009) as our training data, which contains 1.28M images. Compared to the original training process

of CLIP, the incremental alignment step is quite lightweight and can be performed on image-only datasets.

**Training Schemes.** The kernel we use is the normalized polynomial kernel of degree 3, which has been commonly adopted by several well-known studies in the literature (Stein et al., 2023; Kang et al., 2023). We also explore different kernel choices in the ablation studies. For DINOv2, we set the hyper-parameter of kernel to $\gamma = 1/dim_{emb}$ and $c = 1$, while for CLIP, they are set as trainable. More detailed settings for each experiment can be found in the Appendix B.1. With two 4090 GPUs, the alignment of ViT-L-14 takes around 30 hours, which is efficient and hardware-friendly compared to the pre-training phase of CLIP.

## 4.2. Improvements on Vision-centric Tasks

We begin with vision-centric tasks, which can be done by CLIP alone, to verify whether alignment with DINOv2 embeddings can enhance the visual representation of CLIP.

**Zero-shot Object Recognition.** We first test the zero-shot accuracy of the aligned CLIP model on standard object recognition benchmarks. We experiments on a diverse benchmarks composed of 12 datasets, including (1) common objects: ImageNet (Deng et al., 2009), CIFAR-10, CIFAR-100 (Krizhevsky et al., 2009), Caltech101 (Fei-Fei et al., 2004); (2) fine-grained objects: OxfordPets (Parkhi et al., 2012), DTD (Cimpoi et al., 2014), FER2013 (Goodfellow et al., 2013); (3) domain-specific applications: PCAM (Veeling et al., 2018), RESISC45 (Cheng et al., 2017), EuroSAT (Helber et al., 2018); and (4) out-of-distribution benchmarks: ImageNet-O (Hendrycks et al., 2021), ImageNet-Sketch (Wang et al., 2019). We adopt the standard CLIP-Benchmark (LAION-AI, 2022) as the pipeline for evaluation. The results are presented in Table 1. For zero-shot object recognition, performance depends on both perceptual ability and compatibility with the text encoder. After aligning with DINOv2 representations, we

*Table 4.* **Results on probing benchmark.** We report macro-averaged recall for both local probing and global probing.

| Vision Encoder | | local | global |
|---|---|---|---|
| ViT-B-16 | w/o align | 44.63 | 52.61 |
| | w/ align | **45.25** | **52.94** |
| ViT-L-14 | w/o align | 46.40 | 54.51 |
| | w/ align | **47.44** | **55.33** |
| ViT-14-336 | w/o align | 46.05 | 55.13 |
| | w/ align | **46.65** | **56.09** |

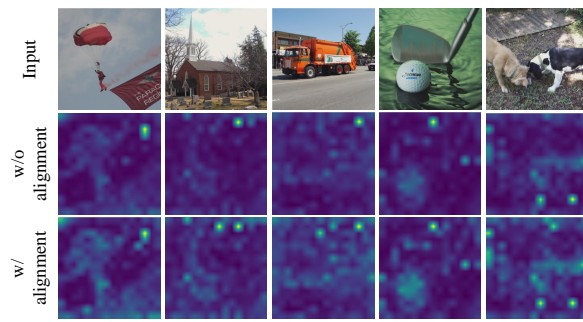

*Figure 2.* **Visualization of CLIP encoder's attention maps.** Attention maps can show more fine-grained details after alignment.

observe improvements across most datasets. Specifically, the average zero-shot accuracy increases by 0.82%, 1.28%, and 1.03% with alignment for ViT-B, ViT-L, and ViT-L-336, respectively. This indicates that our proposed alignment enhances visual representation while maintaining compatibility with the CLIP text encoder. The improvements are particularly notable for images with low resolution (e.g., CIFAR) or tiny objects (e.g., EuroSAT), highlighting enhanced fine-grained perception capabilities. Through experiments with different CLIP backbones, we find that these improvements are architecture-agnostic, consistently boosting classification accuracy. Furthermore, the enhancements hold for both the ImageNet dataset and zero-shot datasets, demonstrating that alignment does not compromise the generalizability of CLIP. Even when fine-tuned on relatively small training data, the model still exhibits strong zero-shot performance on out-of-distribution datasets.

To demonstrate the superiority of the proposed kernel-based alignment, we compare our method with two straightforward baselines: (1) a linear projector trained to map the DINOv2 representation into the CLIP representation space, and (2) replacing kernel-based alignment with feature-based alignment (details available in Appendix B.2). However, we observe limited improvements or even significant drops in zero-shot accuracy. The representation spaces of DINOv2 and CLIP can vary significantly, making it challenging to directly align them while preserving compatibility with the text embeddings. In contrast, our kernel-based alignment focuses on aligning the relative relationships among samples, which helps maintain the macro-structure of the feature space. We also compare our approach with DIVA (Wang et al., 2025), and the results indicate that our method achieves better zero-shot performance. Furthermore, DIVA and our proposed alignment are orthogonal, they can be combined for complementary benefits.

In Appendix C.2, we extend our framework to three additional models that incorporate an intermediate embedding layer linking modalities, albeit with implementations differing from CLIP: SigLIP (Zhai et al., 2023), DFN (Fang et al., 2023), and MetaCLIP (Xu et al., 2024). We also explore replacing DINOv2 with other vision models, such as MLCD (An et al., 2024). The results show improved zero-shot accuracy for all these pairs, demonstrating the generalizability of the proposed framework.

**Image-to-text and Text-to-image Retrievals.** To further demonstrate that the proposed alignment framework does not compromise the image-text alignment or the generalizability of CLIP, we present zero-shot image-to-text and text-to-image retrieval performance in Table 2. These experiments were conducted on both the Flicker30K (Young et al., 2014) and MSCOCO (Chen et al., 2015) datasets. The results indicate that CLIP maintains strong performance in both retrieval tasks even after alignment. This reinforces our assertion that the image-text alignment is effectively preserved, despite the improvements in visual capability.

**Counting, Spatial Reasoning, and Caption Recognition.** As highlighted in the literature (Tong et al., 2024), the CLIP encoder often struggles with specific tasks, including counting, spatial relationship inference, and caption recognition, which serves as a major motivation for our work. To investigate how alignment with DINOv2 can alleviate these issues, we conduct experiments on four related benchmarks: (1) SVHN (Netzer et al., 2011), which composed of natural scene images with digits and numbers; (2) GTSRB (Stallkamp et al., 2012), a datasets for recognition of German traffic sign; (3) CLEVR Distance (Johnson et al., 2017), which composed of images with multiple objects and the task is to determine the distance between the closest objects; and (4) CLEVR Counts, a benchmark that requires the model to count the number of objects within the images. We again follow the evaluation protocol of CLIP-Benchmark, which formulates all four benchmarks as classification tasks. We assess the improvements through both zero-shot classification and linear probing. The results, presented in Table 3, indicate that alignment significantly enhances CLIP's performance on text or spatial-related tasks, particularly after linear probing. For example, when using ViT-L-14, alignment improves the average accuracy by 1.58% for zero-shot classi-

*Table 5.* **Evaluation of LLaVA on vision-language benchmarks.** The performance can be improved by using the CLIP after alignment as the visual encoder, and it can be further improved through parameter-efficient fine-tuning (peft) of the LLM.

| | Vision Encoder | AI2D | text-ocr | text-pure | pope | tallyQA | vsr | RefCOCOg | RefCOCO+ | RefCOCO | vqa-v2 | llava-wild | llava-coco | Average |
|---|---|---|---|---|---|---|---|---|---|---|---|---|---|---|
| **ViT-L** | baseline | 52.91 | 55.37 | 41.16 | 84.64 | 60.39 | 51.47 | 13.89 | 12.73 | 15.16 | 75.07 | 44.13 | 75.07 | 48.50 |
| | +align | 53.66 | **55.70** | 41.53 | **84.92** | **60.39** | 51.50 | 14.15 | 12.89 | 15.37 | **75.10** | **44.67** | **76.60** | 48.87 |
| | +peft | 53.63 | 54.54 | 41.82 | 84.35 | 57.19 | 53.93 | 56.06 | 52.74 | 61.27 | 74.71 | 44.30 | 64.84 | 58.28 |
| | +both | **54.79** | 55.03 | **42.13** | 84.37 | 59.87 | **58.02** | **56.47** | **52.82** | **62.25** | 74.87 | 43.70 | 65.33 | **59.14** |
| **ViT-L-336** | baseline | **53.27** | 57.82 | 48.63 | 85.45 | 61.45 | 51.47 | 55.19 | 51.97 | 60.49 | **76.40** | 42.97 | 75.60 | 60.06 |
| | +align | 52.95 | 57.94 | 48.53 | 85.57 | 61.25 | 51.47 | 55.66 | 52.78 | 60.54 | 76.34 | **48.40** | **79.07** | 60.88 |
| | +peft | 52.69 | 57.79 | 48.50 | 85.41 | 61.91 | 54.14 | 56.76 | 53.81 | 61.98 | 75.88 | 44.13 | 69.13 | 60.18 |
| | +both | 52.78 | **58.18** | **48.87** | **85.70** | **62.21** | **56.22** | **57.68** | **55.74** | **63.37** | 75.93 | 44.90 | 78.83 | **61.70** |

fication and 5.51% for linear probing. Compared to common object recognition, tasks like caption recognition, counting, and spatial reasoning rely more heavily on the quality of the visual representation. This is where vision-centric models such as DINOv2 significantly outperform CLIP. After alignment, the CLIP representation demonstrates enhanced visual capabilities, resulting in better performance on these tasks. Additionally, we include results from MMVP-VLM (Tong et al., 2024) in Appendix C.4, which provides insights from a more diverse but relatively smaller benchmark.

**Localization Ability.** We further evaluate the localization ability of CLIP visual encoders w/wo alignment. Specifically, we utilize the probing benchmark introduced by Covert et al. (2025). This evaluation involves freezing the visual encoder and training a classification head to predict the union of labels for each patch (local probing) or the entire image (global probing) using a binary cross-entropy loss. The experiments are conducted on the MSCOCO dataset (Lin et al., 2014). Following the original setup, we report the macro-averaged recall to account for class imbalances. The results, shown in Table 4, reveal that after alignment with DINOv2, the CLIP visual encoder demonstrates improved localization ability, achieving higher recall for both local and global probing. This enhancement can be attributed to the strong perceptual capabilities of the DINOv2 encoder, which are transferred to CLIP encoder via alignment. Furthermore, the improvements are consistent across different CLIP encoder architectures. While the gains may not be as significant compared to those reported by Covert et al. (2025), their approach disrupts the connection with text representations, leading to a loss of CLIP's zero-shot capability. In contrast, our method enhances performance without compromising the textual alignment.

We visualize the attention maps from the penultimate layer of the CLIP visual encoder for several examples in Fig 2.

The attention maps are generally similar w/wo alignment, as the regularization term prevents significant changes in the model's weights. However, we still observe that the attention maps after alignment are sharper and highlight more fine-grained features. This indicates that the aligned model is better at capturing details from the input image, resulting in improved localization ability.

### 4.3. Improvements on MLLMs

We then show that the enhancements in visual representation can be transferred to downstream tasks. Specifically, we replace the vision encoder of several MLLMs with the aligned CLIP encoder and evaluate improvements on relevant benchmarks. The LLM component can either be entirely frozen or fine-tuned. Importantly, our alignment maintains the similarity of visual representations before and after the process, eliminating the need for an additional vision-language alignment step. As a result, fine-tuning can be conducted efficiently using techniques such as LoRA (Hu et al., 2022).

**Evaluation on LLaVA.** We first conduct experiments on LLaVA-1.5-7B (Liu et al., 2024). We utilize diverse benchmarks which comprises of multiple tasks including (1) open-ended visual question answering: VQAv2 (Goyal et al., 2017) and TextVQA (Singh et al., 2019); (2) localization: RefCOCO, RefCOCO+, RefCOCOg (Kazemzadeh et al., 2014; Yu et al., 2016); and (3) closed-set prediction: VSR (Liu et al., 2023), TallyQA (Acharya et al., 2019), POPE (Li et al., 2023b), and AI2D (Kembhavi et al., 2016). We also perform GPT-aided evaluation, LLaVA-bench (Liu et al., 2024). We replace the visual encoder with our aligned version and experiment with both ViT-L-14 and ViT-L-14-336 versions of CLIP encoder. To better illustrate the enhancements, we further fine-tune the LLM component of the models. We skip the feature alignment step and conduct only visual instruction tuning using LoRA. For the training data,

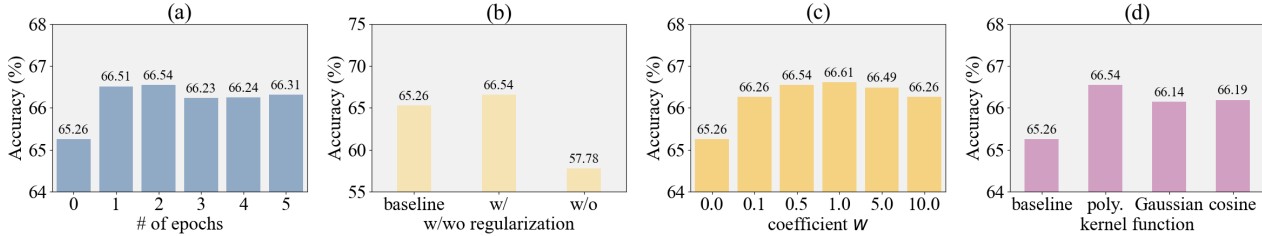

*Figure 3.* **Ablation studies.** Average zero-shot accuracy across 11 datasets are reported as evaluation metrics: (a) Effects of training epochs; (b) Effects of the regularization term; (c) Effects of the coefficients of alignment; (d) Effects of different kernel functions.

*Table 6.* **Evaluation results on OpenFlamingo across seven vision-language datasets using 0, 4, and 16 in-context examples.** The performance can be improved by using the CLIP visual encoder after alignment as the visual encoder, without fine-tuning the LLM part. We report CIDEr for COCO and Flicker-30K, ROC AUC for HatefulMemes, and VQA accuracy for the rest.

| Shots | Vision Encoder | COCO | Flickr-30K | VQAv2 | OK-VQA | TextVQA | VizWiz | HatefulMemes | Average |
|---|---|---|---|---|---|---|---|---|---|
| 0 | CLIP | 74.40 | 54.15 | 43.54 | 28.72 | 25.42 | 17.80 | 51.03 | 42.15 |
|  | +align | **76.01** | **54.42** | **43.71** | **29.04** | **25.92** | **17.91** | **52.86** | **42.84** |
| 4 | CLIP | 82.56 | 58.72 | 45.40 | 31.81 | 29.03 | 23.05 | 48.24 | 45.54 |
|  | +align | **83.95** | **60.03** | **45.57** | **31.82** | **29.43** | **23.12** | **49.22** | **46.16** |
| 16 | CLIP | 90.42 | 62.68 | 45.59 | 32.12 | 29.83 | 35.22 | **50.62** | 49.45 |
|  | +align | **90.89** | **63.85** | **45.73** | **32.43** | **30.33** | **35.74** | 48.85 | **49.69** |

we utilize the LLaVA-1.5 data mixture (Liu et al., 2024), which contains 665k examples and is the tuning dataset for the original LLaVA. We also report metrics from performing the same fine-tuning on the original model to control for the effects of fine-tuning. The results, shown in Table 5, indicate improvements even when the vision encoder is simply replaced with our aligned version, without fine-tuning the LLM component. This suggests that the alignment lead to improvement while preserving the compatibility of CLIP with the LLM component. The improvements are further amplified when we fine-tune the LLM component, resulting in an average score increase of 1.66%, for model with ViT-L-14-336 as the visual encoder. This is particularly evident in the localization benchmarks, with 3.05% improvement on average. Furthermore, the enhancements cannot be solely attributed to LLM fine-tuning, as fine-tuning LLM alone does not yield such great gains. These experiments demonstrate that our proposed alignment enhances visual capabilities, especially fine-grained perception, and these improvements can be effectively transferred to downstream applications.

**Evaluation on OpenFlamingo.** We extend the evaluation to another popular MLLM, OpenFlamingo (Awadalla et al., 2023). We follow the evaluation pipeline of the original paper, which test the in-context-learning ability of the MLLMs in several VQA benchmarks, including COCO (Chen et al., 2015), Flicker-30K (Young et al., 2014), VQAv2 (Goyal et al., 2017), OK-VQA (Marino et al., 2019), TextVQA (Singh et al., 2019), VizWiz (Gurari et al., 2018),

and HatefulMemes (Kiela et al., 2020). For each dataset, we sample a few in-context demonstrations from the training split uniformly at random, and prompt the model to give answers to the test samples. We use the OpenFlamingo-3B (Instruct) version of the model. The results are shown in Table 6. For all experiments, we replace the visual encoder to our aligned version, without fine-tuning the LLM part. We discover the alignment improves most of the metrics, showing the generalization of our proposed framework.

### 4.4. Ablation Studies

We study the impact of various design choices through ablation studies, focusing on factors such as training epochs, loss function design, and the kernel functions utilized. Zero-shot object recognition accuracy serves as our evaluation metric, with results illustrated in Fig.3. Key takeaways include: (1) the method demonstrates optimal performance with a very short training duration; (2) the regularization term is crucial for preserving visual-language alignment; (3) the performance remains stable regardless of the coefficient $w$; and (4) different kernels all contribute positively, with the polynomial kernel yielding the best results. A more comprehensive analysis can be found in Appendix C.7.

## 5. Limitations & Conclusion

In this study, we introduce a novel kernel-based embedding alignment strategy designed to align two sets of embeddings.

Our evaluations on vision-centric tasks and MLLM benchmarks demonstrate that this alignment enhances the visual capabilities of CLIP, with benefits that inherit to downstream applications. Unlike most existing research which enhances the visual encoder but requires re-tuning of the LLM, our approach opens up a new research avenue to facilitate improvements through lightweight adaptation, making these advancements more accessible to a broader community.

Due to the lack of computational resources, we align the embedding on a relatively small-scale dataset, and only evaluate the performance on small MLLMs. It can be left as a future work to conduct the fine-tuning on a larger scale datasets (Gadre et al., 2024) and verify the effects on MLLMs with larger size (e.g., 70B). Another limitation is that this work focuses on the alignment between CLIP and DINOv2. There are MLLMs that utilize vision encoders other than CLIP (Chen et al., 2024), and other of vision-centric models (Bao et al., 2022) available. We would extend the current pipeline to help align other embeddings pairs and bring improvements to more MLLMs.

## Acknowledgements

The work of Farzan Farnia is partially supported by a grant from the Research Grants Council of the Hong Kong Special Administrative Region, China, Project 14209920, and is partially supported by CUHK Direct Research Grants with CUHK Project No. 4055164 and 4937054. Also, this work is supported in part by a grant from the Research Grants Council of the Hong Kong Special Administrative Region, China (Project No. T45-401/22-N). Finally, the authors would like to thank the anonymous reviewers for their insightful suggestions and feedback.

## Impact Statement

Our work enhances the capabilities of multi-modal large language models by improving visual perception through the alignment of visual representations. As shown in the experiments, we improve the performance on both general-purpose computer vision benchmarks and domain-specific applications including remote sensing and medical diagnosis. Therefore, Our method has the potential to significantly advance applications in areas such as autonomous systems and assistive technologies. This progress can lead to more accurate and context-aware AI systems, ultimately benefiting industries like healthcare, transportation, and education.

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

# A. Proofs of Propositions

## A.1. Proof of Proposition 3.1.

This is a conclusion derived from Vector Bernstein Inequality (Gross, 2011; Kohler & Lucchi, 2017), which shows that if $X_1, \cdots, X_M$ are independent vector-valued random variables with common dimension $d$ and that each one is centered, uniformly bounded and also the variance is bounded above:

$$\mathbb{E}[x_m] = 0 \text{ and } \|X_m\|_2 \leq \mu \text{ as well as } \mathbb{E}[\|X_m\|^2] \leq \sigma^2. \tag{7}$$

Let

$$Z = \frac{1}{M} \sum_{m=1}^{M} X_m, \tag{8}$$

then we have for $0 < \epsilon < \sigma^2/\mu$,

$$P(\|Z\|_2 \geq \epsilon) \leq \exp\left(-\frac{M\epsilon^2}{8\sigma^2} + \frac{1}{4}\right). \tag{9}$$

By plugging in $X_m$ with $t(\theta; I_{m_1}, I_{m_2}) - \mathbb{E}[t(\theta)]$, we have

$$\mathop{\mathbb{E}}_{I_{m_1}, I_{m_2} \sim \mathcal{D}_{\text{data}}} [t(\theta; I_{m_1}, I_{m_2})] - \mathbb{E}[t(\theta)] = 0, \tag{10}$$

$$\|t(\theta; I_{m_1}, I_{m_2}) - \mathbb{E}[t(\theta)]\|_2 \leq 8L, \tag{11}$$

$$\mathbb{E}[\|t(\theta; I_{m_1}, I_{m_2}) - \mathbb{E}[t(\theta)]\|_2^2] \leq 64L^2. \tag{12}$$

Therefore, we have:

$$P\left(\left\|\frac{1}{M} \sum_{m=1}^{M} t(\theta; I_{m_1}, I_{m_2}) - \mathbb{E}[t(\theta)]\right\|_2 \geq \epsilon\right) \leq \exp(-\frac{M\epsilon^2}{512L^2} + \frac{1}{4}). \tag{13}$$

## A.2. Proof of Proposition 3.2.

The proposition and the proof are adapted from (Schlarmann et al., 2024). We have

$$|\cos(f_{\theta_0}(I), g(T)) - \cos(f_\theta(I), g(T))| = \left|\left\langle \frac{g(T)}{\|g(T)\|_2}, \frac{f_{\theta_0}(I)}{\|f_{\theta_0}(I)\|_2} - \frac{f_\theta(I)}{\|f_\theta(I)\|_2}\right\rangle\right| \leq \left\|\frac{f_{\theta_0}(I)}{\|f_{\theta_0}(I)\|_2} - \frac{f_\theta(I)}{\|f_\theta(I)\|_2}\right\|_2.$$

For which we can get the two upper bounds:

$$\left\|\frac{f_{\theta_0}(I)}{\|f_{\theta_0}(I)\|_2} - \frac{f_\theta(I)}{\|f_\theta(I)\|_2}\right\|_2 \leq \frac{1}{\|f_\theta(I)\|_2} \left[|\|f_\theta(I)\|_2 - \|f_{\theta_0}(I)\|_2| + \|f_{\theta_0}(I) - f_\theta(I)\|_2\right], \tag{14}$$

$$\left\|\frac{f_{\theta_0}(I)}{\|f_{\theta_0}(I)\|_2} - \frac{f_\theta(I)}{\|f_\theta(I)\|_2}\right\|_2 \leq \frac{1}{\|f_{\theta_0}(I)\|_2} \left[|\|f_\theta(I)\|_2 - \|f_{\theta_0}(I)\|_2| + \|f_{\theta_0}(I) - f_\theta(I)\|_2\right]. \tag{15}$$

According the triangle inequality:

$$|\|f_{\theta_0}(I)\|_2 - \|f_\theta(I)\|_2| \leq \|f_{\theta_0}(I) - f_\theta(I)\|_2, \tag{16}$$

therefore, the upper bound holds:

$$|\cos(f_{\theta_0}(I), g(T)) - \cos(f_\theta(I), g(T))| \leq \min\left(\frac{2}{\|f_\theta(I)\|_2}, \frac{2}{\|f_{\theta_0}(I)\|_2}\right) \|f_{\theta_0}(I) - f_\theta(I)\|_2. \tag{17}$$

**Remarks.** This proposition demonstrates that minimizing the $L_2$ distance between the visual representations before and after the alignment fine-tuning help directly preserve the visual-text alignment. This is different from previous work such as Xuhong et al. (2018); Li et al. (2020), which penalizes the $L_2$-norm of parameter changes during the fine-tuning phase. While their method effectively prevents overfitting to the target domain in transfer learning, our direct feature-based penalty measure aims to maintain zero-shot compatibility while enhancing fine-grained visual capabilities.

# B. More Implementation Details

### B.1. Hyper-parameters Setup.

Here we provide more information on the implementation details. Specifically, the hyper-parameters used for the alignment fine-tuning is listed in Table 7. All the experiments are conducted on NVIDIA GeForce RTX 4090 GPUs.

*Table 7.* **Detailed hyper-parameter setups.**

| Hyper-parameters | ViT-B-16 | ViT-L-14 | ViT-L-14-336 |
|---|---|---|---|
| coefficient $w$ | 0.5 | 0.5 | 1.0 |
| number of GPUs | 2 | 2 | 4 |
| batch size | 128 | 64 | 32 |
| training epochs | 2 | 2 | 4 |
| optimizer | AdamW (Loshchilov, 2017) | | |
| weight decay | 1e-4 | | |
| $\beta$ | (0.9, 0.999) | | |
| learning rate | 1e-5 | | |
| scheduler | CosineAnnealingLR | | |
| warm-up steps | 1400 | 2800 | 5600 |

### B.2. Illustration on Feature-based Alignment.

A vanilla way to align the CLIP embedding $f_\theta(I_i)$ with the embedding of the target model $g(I_i)$ is to minimize the $L_2$ distance between $f_\theta(I_i)$ and $g(I_i)$ subject to a linear transformation $\mathbf{R} \in \mathbb{R}^{d \times d'}$:

$$\min_{\theta, \mathbf{R}} \mathbb{E}_{I_i \sim \mathcal{D}_{\text{train}}} [w \| f_\theta(I_i) - \mathbf{R}g(I_i) \|_2^2 + \| f_\theta(I_i) - f_{\theta_0}(I_i) \|_2^2]. \tag{18}$$

We call this method feature-based embedding alignment. This approach also make sense and lead to certain improvements compared with the original visual embeddings of CLIP. However, the representation space of CLIP and the target model often vary significantly, which can hardly be adjusted via linear transformation. As a result, it is an sub-optimal solution. On the contrary, our kernel-based embedding alignment provide a more flexible solution. Firstly, comparing with absolute position within the feature space, the relative position among samples is more important. We would like the embeddings of two samples with similar visual characteristics to be close to each other. Using kernel function as supervision can directive promote this effects. Secondly, kernel function measures the similarity in the high-dimensional kernel space. Alignment of embeddings within the kernel space will not lead to drastic changes of the embeddings in the original feature space, making the image embeddings after alignment still compatible with text encoder and downstream modules. Moreover, our kernel-based alignment can actually be interpreted as aligning the representations in the kernel space. The feature transformation $\phi$ for common kernels are usually non-linear. It renders more flexibility compared to the feature-based alignment which only use linear transformation.

### B.3. Implementation Details for LLM's Fine-tuning.

In this section, we elaborate on the implementation details of the LLM fine-tuning of LLaVA, which we used to further demonstrate the enhancement of the vision encoder with alignment. We employ the official implementation from LLaVA for LoRA fine-tuning. The training is conducted on a mixture of LLaVA-1.5 data for one epoch, using the following LoRA configuration: $r = 128$ and $\alpha = 256$. The training is executed in bf16 format across four NVIDIA GeForce RTX 4090 GPUs, with a batch size of 1 per device. To address the small batch size, we apply a gradient accumulation step of 32. The optimizer used is AdamW (Loshchilov, 2017), set with a learning rate of 2e-4 and a weight decay of 0.

# C. Additional Experimental Results

## C.1. Loss Curve for the Alignment Term.

We visualize the alignment loss (with moving average smoothing) during the fine-tuning phase in Fig. 4. We discover the loss term is stably decreasing during the fine-tuning phase, showing that the alignment term is indeed effectively optimized to improve the similarity between CLIP and DINOv2 representations.

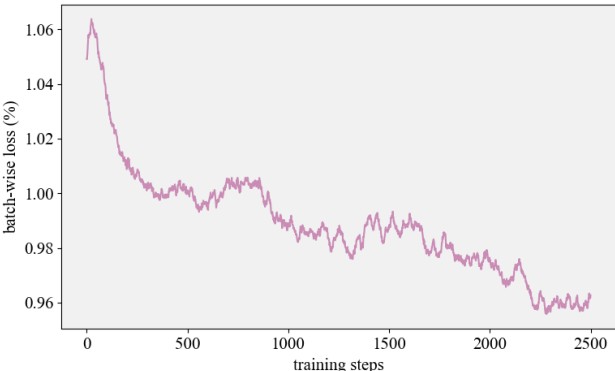

*Figure 4.* **Visualization of the alignment loss curve.** The alignment loss is stably decreasing during the alignment fine-tuning phase.

## C.2. Results for VLMs other than CLIP.

To evaluate the generalization of our proposed framework, we conducted experiments on three additional models that feature an intermediate embedding layer connecting modalities, but with implementations distinct from CLIP: SigLIP (Zhai et al., 2023), DFN (Fang et al., 2023), and MetaCLIP (Xu et al., 2024). For all the VLMs, we utilized the ViT-L-14 version, maintaining the same implementation details as those for CLIP ViT-L-14. We assessed zero-shot accuracy across 12 datasets as our evaluation metric, with results presented in Table 8. The findings indicate that our alignment fine-tuning enhances zero-shot performance for all three models, yielding average increases of 0.45%, 0.48%, and 0.87% respectively. These results highlight the generalizability of our framework, demonstrating that the alignment is applicable to VLMs beyond CLIP. Despite ongoing advancements in CLIP research, our proposed alignment remains adaptable and beneficial for these more advanced VLMs.

*Table 8.* **Accuracy evaluation on zero-shot image classification benchmarks of vision-language models w/wo alignment**. We experiments with ViT-L-14 version of SigLIP (Zhai et al., 2023), DFN (Fang et al., 2023), and MetaCLIP (Xu et al., 2024).

| Vision Encoder | ImageNet | Zero-shot Datasets | | | | | | | | | | | Average Zero-shot |
| | | CIFAR10 | CIFAR100 | CalTech | FER | OxfordPets | DTD | RESISC | EuroSAT | PCAM | ImageNet-S | ImageNet-O | |
| --- | --- | --- | --- | --- | --- | --- | --- | --- | --- | --- | --- | --- | --- |
| SigLip | 82.02 | 96.87 | 84.27 | 86.01 | 49.90 | 95.31 | 71.06 | 72.59 | 62.65 | 50.55 | 74.02 | 29.60 | 70.26 |
| +align | **82.18** | 96.88 | 84.89 | 86.11 | 49.16 | 95.42 | 71.06 | 72.84 | 65.54 | 50.11 | 73.92 | 31.90 | **70.71** |
| DFN | **81.50** | 98.32 | 88.15 | 85.57 | 42.70 | 95.58 | 66.06 | 73.25 | 64.94 | 63.18 | 68.33 | 39.30 | 71.40 |
| +align | 81.28 | 98.12 | 87.82 | 85.49 | 44.27 | 95.39 | 66.70 | 73.76 | 65.44 | 62.90 | 68.40 | 42.35 | **71.88** |
| MetaCLIP | 75.58 | 95.67 | 77.72 | 85.74 | 37.96 | 93.81 | 62.34 | 68.52 | 60.41 | 70.35 | 65.05 | 28.90 | 67.86 |
| +align | **76.60** | 96.42 | 80.11 | 85.49 | 40.39 | 93.68 | 62.82 | 69.25 | 60.35 | 69.76 | 65.12 | 32.65 | **68.73** |

## C.3. Results for Other Target Models.

We also investigate the potential of replacing DINOv2 with alternative vision encoders to demonstrate the flexibility of our proposed framework. Specifically, we employed MLCD (An et al., 2024) as the target model, with results displayed in Table 9. The findings suggest that switching the target model from DINOv2 to other vision encoders can lead to significant improvements, highlighting the generalizability of our framework. As vision-centric self-supervised learning progresses and larger image datasets are developed, we expect to see the emergence of even more powerful vision encoders in the future. Our research establishes a foundation for utilizing these advanced encoders to enhance the performance of vision-language models and potentially benefit downstream MLLMs.

*Table 9.* **Accuracy evaluation on zero-shot image classification benchmarks of CLIP models w/wo alignment**. We use MLCD instead of DINOv2 as the target model for alignment. The results show the alignment still show improvement when using different target model.

| Vision Encoder | ImageNet | Zero-shot Datasets | | | | | | | | | | | Average Zero-shot |
| | | CIFAR10 | CIFAR100 | CalTech | FER | OxfordPets | DTD | RESISC | EuroSAT | PCAM | ImageNet-S | ImageNet-O | |
|---|---|---|---|---|---|---|---|---|---|---|---|---|---|
| ViT-L-14 | 74.90 | 95.20 | 71.08 | 83.30 | 50.00 | 93.21 | 55.21 | 63.35 | 62.65 | 52.00 | 59.59 | 32.25 | 65.26 |
| +align | **75.32** | 96.03 | 74.90 | 83.86 | 49.67 | 93.43 | 55.59 | 64.62 | 63.44 | 52.28 | 59.99 | 34.70 | **66.26** |
| ViT-L-14-336 | 75.82 | 94.49 | 71.11 | 83.42 | 49.04 | 93.70 | 55.64 | 63.73 | 61.46 | 60.68 | 61.03 | 32.75 | 66.10 |
| +align | **76.18** | 95.90 | 76.70 | 83.53 | 49.00 | 94.06 | 56.60 | 63.90 | 61.54 | 60.66 | 61.57 | 36.30 | **67.25** |

## C.4. Results on MMVP-VLM benchmarks.

We conduct experiments on the MMVP-VLM benchmarks (Tong et al., 2024). The benchmark consists of "CLIP-blind pair" images, which the CLIP vision encoder recognizes as similar, despite notable visual differences. It is designed to test whether the CLIP model can differentiate between these pairs. The pairs encompass a range of visual patterns, including orientation and direction, the presence of specific features, state and condition, quantity and count, positional and relational context, color and appearance, structural and physical characteristics, text, and viewpoint and perspective. The results are detailed in Table 10. Findings indicate that the proposed alignment improves accuracy by 2.97%, 2.96%, and 2.22% for three variants of CLIP models. These results demonstrate that aligning with DINOv2 helps address the limitations of CLIP in these challenging cases.

*Table 10.* **Performance of CLIP on various visual patterns of MMVP-VLM benchmark.** Alignment with DINOv2 embeddings greatly overcomes CLIP's original shortcomings in terms of perceiving visual details. Symbols for visual patterns as (Tong et al., 2024) are inherited: 🧭: Orientation and Direction, 🔍: Presence of Specific Features, 🔄: State and Condition, 🔢: Quantity and Count, 📍: Positional and Relational Context, 🎨: Color and Appearance, ⚙: Structural and Physical Characteristics, **A**: Texts, 📷: Viewpoint and Perspective.

| Vision Encoder | | 🧭 | 🔍 | 🔄 | 🔢 | 📍 | 🎨 | ⚙ | A | 📷 | Average |
|---|---|---|---|---|---|---|---|---|---|---|---|
| ViT-B-16 | w/o align | 6.67 | 0.00 | 26.67 | 13.33 | 13.3 | 20.00 | 13.33 | 0.00 | 20.00 | 12.59 |
| | w/ align | 6.67 | 0.00 | 20.00 | 13.33 | 20.00 | 26.67 | 26.67 | 0.00 | 26.67 | **15.56** |
| ViT-L-14 | w/o align | 6.67 | 13.33 | 20.00 | 13.33 | 6.67 | 53.33 | 26.67 | 6.67 | 13.33 | 17.78 |
| | w/ align | 13.33 | 20.00 | 20.00 | 20.00 | 6.67 | 53.33 | 33.33 | 13.33 | 20.00 | **20.74** |
| ViT-L-14-336 | w/o align | 0.00 | 20.00 | 40.00 | 20.00 | 6.67 | 20.00 | 33.33 | 0.00 | 33.33 | 19.26 |
| | w/ align | 6.67 | 26.67 | 40.00 | 13.33 | 6.67 | 40.00 | 26.67 | 13.33 | 20.00 | **21.48** |

## C.5. Results on MMVP benchmarks.

We also conducted experiments on the MMVP benchmarks (Tong et al., 2024), which are similar to the MMVP-VLM benchmark but evaluate through VQA on MLLMs. For this experiment, we used LLaVA-1.5-7B with CLIP ViT-L-14-336 as the visual encoder. The results are shown in Table 11. We found that the proposed alignment framework significantly

enhances the performance of LLaVA, leading to a marked reduction in error rates for these challenging cases. Improvements were observed simply by replacing the vision encoder with the aligned version, and the gains were even more pronounced when we fine-tuned the LLM component as well. These results further confirm that the enhanced visual capabilities of CLIP can be effectively transferred to downstream MLLMs.

*Table 11.* **Performance of LLaVA on MMVP benchmark.** The performance improves when we replace the visual encoder with the aligned version. This indicates that alignment enhances visual capabilities, and this effect can be carried over to downstream tasks.

| Vision Encoder | CLIP | CLIP+align | CLIP+align+peft |
|---|---|---|---|
| Accuracy (%) | 23.33 | 24.67 | **34.00** |

## C.6. Additional Baselines

In this section, we compare our method with two additional baselines, which also entail integrating knowledge from multiple experts to build a stronger visual encoder.

**Comparison with AM-RADIO** (Ranzinger et al., 2024). AM-RADIO distills knowledge from multiple teacher models into a student model, delivering strong performance across various downstream tasks. However, the original AM-RADIO model is trained on DataComp-1B, a dataset consisting of 13 billion samples. This training process requires computational resources comparable to the pre-training of CLIP, making it highly resource-intensive. In contrast, our proposed kernel-based alignment method achieves strong accuracy and generalizability through fine-tuning on relatively small datasets, such as ImageNet-1k, over just a few epochs. To provide a clearer comparison, we follow the AM-RADIO training protocol and train a model on ImageNet-1k using CLIP and DINOv2 as teacher models (both employing ViT-L-14 backbones), with another ViT-L-14 as the student model. We then evaluate its zero-shot classification performance, as shown in Table 12. The results reveal that AM-RADIO struggles to generalize to out-of-distribution datasets under this setup. In contrast, our alignment method achieves both strong performance and superior generalizability, even with limited data and computational resources.

*Table 12.* **Comparison between our alignment fine-tuning and AM-RADIO.** Our method represent better performance when the training data is limited.

| Vision Encoder | ImageNet | Zero-shot Datasets | | | | | | | | | | | Average Zero-shot |
|---|---|---|---|---|---|---|---|---|---|---|---|---|---|
| | | *CIFAR10* | *CIFAR100* | *CalTech* | *FER* | *OxfordPets* | *DTD* | *RESISC* | *EuroSAT* | *PCAM* | *ImageNet-S* | *ImageNet-O* | |
| AM-RADIO | 75.38 | 95.04 | 71.48 | 78.47 | 25.66 | 83.62 | 45.37 | 38.48 | 31.35 | 50.02 | 51.34 | 69.45 | 58.21 |
| Ours | 75.52 | 96.27 | 77.04 | 84.32 | 50.31 | 93.40 | 55.74 | 63.83 | 63.83 | 52.68 | 59.67 | 34.90 | 66.54 |

**Comparison with Additive-MoF** (Tong et al., 2024). We also compare our method with Additive-MoF, a straightforward approach to combining visual representations from different visual encoders for MLLM applications. This comparison is conducted on vision-language benchmarks using the LLaVA implementation, with the results presented in Table 13. While Additive-MoF achieves better performance in certain tasks, such as object detection, it performs suboptimally in others, such as open-ended visual question answering. As highlighted in the original paper (Tong et al., 2024), simply combining CLIP and DINOv2 features involves an inherent trade-off between visual grounding accuracy and instruction-following capability. In contrast, our alignment fine-tuning approach enhances visual capabilities while maintaining strong alignment with the textual embedding space. Additionally, our framework does not require full re-tuning of the LLM component, making it particularly beneficial in resource-limited settings.

*Table 13.* **Comparison between Additive-MoF and ours on vision-language benchmarks.** Our alignment fine-tuning approach achieves enhanced visual capabilities without compromising alignment to the textual embedding space

| Vision Encoder | AI2D | text-ocr | text-pure | pope | tallyQA | vsr | RefCOCOg | RefCOCO+ | RefCOCO | vqa-v2 | llava-wild | llava-coco | Average |
|---|---|---|---|---|---|---|---|---|---|---|---|---|---|
| Additive-MoF | 51.36 | 45.93 | 18.66 | 86.58 | 63.18 | 51.47 | 68.44 | 65.34 | 69.69 | 75.12 | 41.54 | 74.50 | 59.32 |
| Ours | 53.95 | 58.89 | 50.51 | 86.05 | 62.14 | 52.55 | 58.68 | 57.74 | 65.79 | 76.97 | 52.20 | 78.90 | 62.86 |

### C.7. Analysis of Ablation Studies

We conduct several ablation studies to examine the effects of several design choices in our framework. The experiments are performed with CLIP ViT-L-14, and average zero-shot accuracy across 11 datasets are used as evaluation metrics.

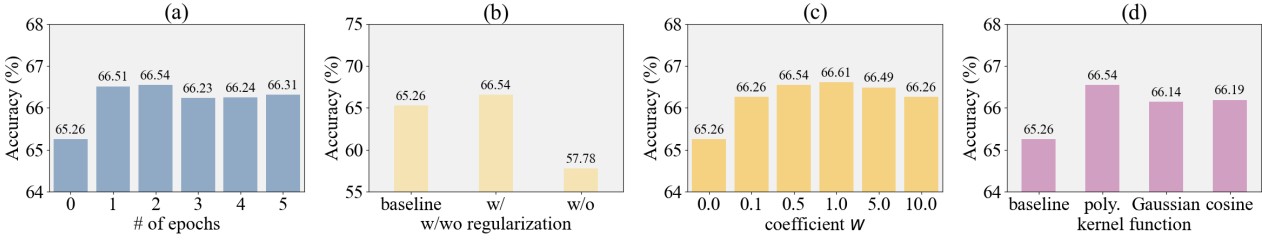

*Figure 5.* **Ablation studies.** Average zero-shot accuracy across 11 datasets are reported as evaluation metrics: (a) Effects of training epochs; (b) Effects of the regularization term; (c) Effects of the coefficients of alignment; (d) Effects of different kernel functions.

**Effects of Training Epochs.** We experiments with fine-tuning for more epochs to see how it affects the performance. The results are shown in Fig. 5 (a). We find that the performance stabilizes after training for 2 epochs. Further training leads to no improvements. This finding highlights the effectiveness of our proposed method, which achieves strong performance with a minimal training time.

**Effects of Regularization.** We also conducted an ablation study on the regularization term in Eq. 6, specifically fine-tuning the CLIP model using only the kernel alignment term. The results are shown in Fig. 5 (b). Without this regularization term, we observe a significant drop in performance, with zero-shot accuracy even worse than that of the original CLIP model. This indicates that the regularization is crucial for maintaining alignment with the text embeddings. Only when both terms are utilized does the alignment phase achieve an effective balance between fine-grained visual capability and compatibility with text semantics.

**Effects of Coefficients.** We then tested various coefficients $w$ for the alignment term in Eq. 6, experimenting with values of 0.1, 0.5, 1.0, 5.0, and 10.0. The results are presented in Fig. 5 (c). We found that performance remains relatively stable across these coefficient changes. In our experiments, a coefficient of 1.0 yielded the best performance, effectively balancing alignment with the DINOv2 representation and compatibility with the text encoder.

**Selection of Kernel Function.** For the main experiment, we employed a normalized polynomial kernel for alignment and also tested the Gaussian kernel and cosine kernel, with results shown in Fig. 5 (d). Our findings indicate that all three kernels lead to performance improvements, with the polynomial kernel achieving the best results. The Gaussian kernel encounters gradient vanishing issues, complicating the optimization process, while the cosine kernel lacks additional parameters to flexibly address differences in feature dimensions and structures, making them sub-optimal compared to the polynomial kernel. One potential avenue for future improvement is to explore more diverse kernel functions or to combine multiple kernel functions as a compositional kernel for alignment.

**Effects of the dataset size.** We investigate how the size of the fine-tuning dataset impacts model performance. To this end,

we perform ablation studies using 25% and 50% of the ImageNet dataset for fine-tuning. The results, presented in Table 14, indicate that our alignment fine-tuning consistently enhances model performance, even when only 25% of the samples are used, demonstrating the method's efficiency. Furthermore, the performance improvement grows as the amount of fine-tuning data increases, highlighting the potential for further gains when larger datasets are utilized.

*Table 14.* **Ablation study in terms of fine-tuning data size.** We conduct the alignment fine-tuning with different proportions of ImageNet data. The model demonstrates consistent performance gain, and the performance improves as more data is incorporated.

| Proportion | ImageNet | Zero-shot Datasets | | | | | | | | | | | Average Zero-shot |
| --- | --- | --- | --- | --- | --- | --- | --- | --- | --- | --- | --- | --- | --- |
| | | CIFAR10 | CIFAR100 | CalTech | FER | OxfordPets | DTD | RESISC | EuroSAT | PCAM | ImageNet-S | ImageNet-O | |
| 0% | 74.90 | 95.20 | 71.08 | 83.30 | 50.00 | 93.21 | 55.21 | 63.35 | 62.65 | 52.00 | 59.59 | 32.25 | 65.26 |
| 25% | 75.08 | 95.84 | 74.13 | 83.89 | 50.59 | 93.40 | 55.32 | 64.33 | 61.54 | 52.02 | 59.96 | 35.30 | 66.03 |
| 50% | 75.34 | 95.87 | 76.05 | 83.91 | 50.29 | 93.59 | 55.90 | 64.41 | 64.00 | 52.53 | 59.98 | 34.75 | 66.48 |
| 100% | 75.52 | 96.27 | 77.04 | 84.32 | 50.31 | 93.40 | 55.74 | 63.83 | 63.83 | 52.68 | 59.67 | 34.90 | 66.54 |

