# OpenReview forum: "Kernel-based Unsupervised Embedding Alignment for Enhanced Visual Representation in Vision-language Models"
_ICML.cc/2025/Conference — ICML 2025 poster_

### Official Review · Reviewer_u8xU · 2025-03-13

**Overall Recommendation:** 3

**Summary:**

In this paper, the author proposed a kernel-based method to tune CLIP models for fine-grained visual information. Specifically, they propose a kernel based objective to align features from the original CLIP and another visual model that is capable of focusing on fine-grained details. Additionally, they introduce a regularization to prevent the tuning deviates from the original too far. Experiments are conducted on visual-centric tasks and MLLM-related tasks. Multiple ablations are also conducted to validate the effectiveness of the proposed method.

**Claims And Evidence:**

Some claims are supported but another important one is not. See below for details

**Essential References Not Discussed:**

See below for details.

**Experimental Designs Or Analyses:**

The experimental design makes sense

**Methods And Evaluation Criteria:**

The evaluation criteria generally makes sense. But additional baselines are required. See below for details.

**Other Comments Or Suggestions:**

Difference between kernel-based objectives and distillation-based objectives (I am just considering this point as an additional opinion not a major weakness.) From the information perspective, the improvement gains mainly come from learning from DINOv2 on the fine-tuning dataset. The kernel-based objective does not introduce additional information to improve the model (same for other distillation-based objectives). The difference between the proposed method and others are just the objective formats. Why or under what circumstances the proposed method is better than others should be further investigated. In my opinion, the fine-tuning dataset could be such a variable. Whether the proposed method is still better than others in a smaller or larger dataset is encouraged to explore.

**Other Strengths And Weaknesses:**

Pros:

1. Experimental analysis is thorough, covering CLIP-based and MLLM-based tasks.

2. Some claims of the proposed method are supported by theory.

Cons:

1. One major claim of the proposed method, in line 88-92 in the introduction, is not well supported. Why aligning features in the kernel space can “preserve the integrity of the original feature space while adjusting the similarities among samples based on their visual details.“ is not explained in the paper. In my understanding, the original feature space will be changed once the encoder is trained (which produces features in the space) regardless of what training objectives are used. In addition, the preservation seems to mainly come from the regularization term instead of the kernel objective in the method. The regularization term, however, is not new and was previously proposed by L2SP [1][2]. More analysis is needed to support the claim.

2. The training loss risks being very small. The polynomial kernel takes a degree of 3 in experiments. The objective in Equation 4 further involves a square operation. I am concerned that the loss from the equation could be very small. It could lead to inefficient training, overflow, or gradient vanishing. In addition, this could lead to very limited choices of kernel functions and their coefficients (the authors found gradient vanishing using gaussian kernels in the supplementary.)

3. Additional baseline is required for vision-centric tasks. The baseline CLIP seems to be directly taken from the pretrained version. However, other baselines including the proposed method are tuned on the ImageNet. This raises concern whether the improvement compared with CLIP is from the fine-tuning dataset or the method itself. The CLIP-ViT fine-tuned on ImageNet should also be involved.

[1] Xuhong L I, Grandvalet Y, Davoine F. Explicit inductive bias for transfer learning with convolutional networks[C]//International conference on machine learning. PMLR, 2018: 2825-2834.

[2] Li X, Grandvalet Y, Davoine F, et al. Transfer learning in computer vision tasks: Remember where you come from[J]. Image and Vision Computing, 2020, 93: 103853.

**Questions For Authors:**

Are there projection heads for CLIP or DINOv2 before they are computed in the kernel? It seems that their original dimensions are the same. What if we want to align two encoders with different dimensions and what will be the impact if applying projection?

**Relation To Broader Scientific Literature:**

The proposed method could have some boarder relations with general visual and multimodal representation learning.

**Theoretical Claims:**

Some claims are theoratically supported.

---

> ### Author Rebuttal · Authors · 2025-03-31
>
> We thank Reviewer u8xU for the thoughtful and constructive feedback on our work. The following is our response to the comments and questions in the review.
> ___
> > **Preserving the original CLIP space in the alignment with DINOv2**
>
> We would like to clarify that our objective function for the alignment task is designed to minimize the changes to the CLIP embedding while aligning it with DINOv2. We agree that we have to make some modifications to the CLIP model to have a higher alignment with the DINOv2. Meanwhile, to minimize the changes to CLIP in alignment, our proposal is to match only the kernel similarity matrices of the CLIP and DINOv2 embeddings, which only focus on the **relative positioning** of the points. In other words, the proposed objective function does not require the absolute position of CLIP embedded point to be identical to that of DINOv2, but the goal is to only obtain similar similarity scores between the two embeddings. In this sense, our alignment method attempts to mitigate the changes to the CLIP model as much as possible.
> ___
> > **Relationship with L2SP**
>
> Our regularization minimizes the $L_2$ difference between visual embeddings of the fine-tuned and original encoders, directly preserving visual-text alignment (as demonstrated in Proposition 3.2). We note that this is different from L2SP that penalizes the $L_2$-norm of parameter changes. While L2SP effectively prevents overfitting to the target domain in transfer learning, our direct feature-based penalty aims to maintain compatibility with the text encoder while enhancing fine-grained visual capabilities. We will include the discussion in the revised paper.
> ___
> > **Clarification on training loss**
>
> The choice of the polynomial kernel function is inspired by the literature of generative model evaluation and the KID metric [[Bińkowski et al., 2018](https://arxiv.org/abs/1801.01401)], which is an established metric to measure the similarity between two sets of representations. In the standard KID evaluation, the kernel function is commonly chosen to be polynomial degree 3. This choice has been commonly adopted by other well-known studies in the literature, e.g. [[Stein et al., 2023](https://arxiv.org/abs/2306.04675)],  [[Kang et al., 2023](https://ieeexplore.ieee.org/abstract/document/10224337)]. We have also drawn the kernel loss in Fig. 4 of the Appendix, which remains at a reasonable magnitude throughout training.
>
> Furthermore, we would like to clarify that the choice of poly-3 kernel function is only to have a reasonable similarity score between **already embedded** samples, and the structure of the kernel similarity matrix is supposed to be governed by the choice of embedding (DINOv2 in our work), not by different kernel functions. Considering this, we think our alignment method is flexible and allows the application of different embedding models. For example, we have already attempted to align with MLCD embedding in Appendix C.3.
> ___
> > **Additional baseline on imagenet fine-tuning**
>
> We thank the reviewer for pointing this out. We agree that comparison with a vanilla ImageNet-fine tuned CLIP (without kernel-based alignment) will shed light on how important the role of kernel-based alignment is in the improvement. To address this comment, we tested the following baseline: fine-tuning the CLIP visual encoder with contrastive loss, where the paired text for each image is “This is a photo of {*class name*}”. While this approach improves ImageNet zero-shot classification, it deteriorates the generalization of the model to other datasets.
>
> |methods|imagenet zero-shot|average zero-shot across 11 dataset|
> |-|-|-|
> |baseline|84.10|57.12|
> |ours|75.52|66.54|
>
> If the reviewer has any specific fine-tuning method in mind, we would be happy to run and report the results during discussion.
> ___
> > **Ablation on the dataset size**
>
> Thank you for the suggestion. We have conducted ablation studies using 25% and 50% of the imagenet for fine-tuning. The results show that our alignment fine-tuning consistently improves the scores even with 25% of the samples. We will include the results and analysis in the revised paper.
> |proportion of data|imagenet zero-shot|average zero-shot across 11 dataset|
> |-|-|-|
> |0%|74.90|65.26|
> |25%|75.08|66.03|
> |50%|75.34|66.48|
> |100%|75.52|66.54|
> ___
> > **Regarding projection heads**
>
> We would like to clarify that we do not use any projection heads when computing the kernel; it is computed directly on the representations produced by the models. Meanwhile, in Table 1,  we also show experiments on aligning two models with different representation dimensions (e.g., CLIP-B and DINOv2-L), where we observe similar improvements.
> ___
> **Reference**
> - Bińkowski et al. "Demystifying MMD GANs." ICLR (2018).
> - Stein et al. "Exposing flaws of generative model evaluation metrics and their unfair treatment of diffusion models." NeurIPS (2023).
> - Kang et al. "StudioGAN: a taxonomy and benchmark of GANs for image synthesis." TPAMI (2023).

---

### Official Review · Reviewer_3gYu · 2025-03-14

**Overall Recommendation:** 3

**Summary:**

The paper aims to enhance the fine-grained perception capability of CLIP by aligning with DINOv2 in the kernel space. It finds that such a practice can resolve the feature space conflict problem when aligning two different forms of features. Improvements are observed in image-text retrieval benchmarks and MLLM benchmarks.

**Claims And Evidence:**

Yes

**Essential References Not Discussed:**

AM-RADIO: Agglomerative Vision Foundation Model, CVPR 2024

**Experimental Designs Or Analyses:**

Yes

**Methods And Evaluation Criteria:**

Yes

**Other Comments Or Suggestions:**

See weaknesses

**Other Strengths And Weaknesses:**

**Strengths**
- The paper is clearly written and easy to follow. The illustrations and tables are also clear.
- The method is well-motivated and can resolve the feature conflict problem of different multiple teacher models.
- The experimental results are comprehensive and support the claims.

**Weaknesses**
- The improvement is slightly marginal compared with the CLIP baseline on MLLM evaluation benchmarks. Morover, the authors does not follow a standard LLaVA training pipeline, e.g., skipping the alignment stage and using LoRA rather than fully fine-tuning. I understand that it is may because the authors are limited by computational resources. But considering the severe fluctuation in MLLMs evaluation results, larger improvements are expected to be observed.
- More methods should be compared under the authors' settings, such as the AM-RADIO models, which use simple feature alignment to distill features of multiple teacher models.

[1] AM-RADIO: Agglomerative Vision Foundation Model, CVPR 2024

**Questions For Authors:**

See weaknesses

**Relation To Broader Scientific Literature:**

Different from prior works, it presents a new method to align with multiple teacher models.

**Theoretical Claims:**

Yes

---

> ### Author Rebuttal · Authors · 2025-03-31
>
> We thank Reviewer 3gYu for the thoughtful and constructive feedback on our work. The following is our response to the comments and questions in the review.
> ___
> > **Significance of the improvements**
>
> We would like to highlight that our alignment fine-tuning achieves consistent improvements on several kinds of vision-centric benchmarks. We improve the performance of CLIP by 1.28%, 1.34%, 5.51%, and 2.96% on zero-shot classification, image-text retrieval, linear probing, and MMVP-VLM benchmark, respectively. For the MLLM benchmarks, we focus on consistent gains across multiple benchmarks and aim to highlight that our kernel-based approach provides a principled and efficient way to enhance model performance, even under resource-constrained settings. Meanwhile, following the suggestions of Reviewer ue7L, we have conducted experiments with standard two-stage training of LLaVA, which shows even greater improvements (under the response for Reviewer ue7L: *The explanation for additional PEFT*). Overall, we think our proposed alignment fine-tuning provides considerable improvements with very low demand for data and computational resources.
> ___
> > **Comparison with AM-RADIO models**
>
> We thank the reviewer for recommending the related work. AM-RADIO [[Ranzinger et al., 2024](https://arxiv.org/abs/2312.06709)] distills knowledge from multiple teachers into a student model, which achieves superior performance on multiple downstream tasks. However, the original AM-RADIO model is trained with DataComp-1B, which is composed of 13B samples. This training process incurs a computational cost equivalent to the pre-training of CLIP, making it resource-intensive. In contrast, our proposed kernel-based alignment focuses on achieving strong accuracy and generalizability through fine-tuning on relatively small datasets, such as ImageNet-1k, for only a few epochs. To provide a fair comparison under our settings, we trained AM-RADIO on ImageNet-1k using CLIP and DINOv2 as teacher models (both with ViT-L-14 as backbones) and another ViT-L-14 as the student model. We evaluated its zero-shot classification performance and observed that AM-RADIO does not generalize well to out-of-distribution datasets under this setup. In contrast, our alignment method demonstrates both strong performance and superior generalizability, even with limited data and computational resources. We will include the results and discussion in the revised version of the paper.
>
>
> | Task  |imagenet|cifar10|cifar100|caltech|fer|pets|dtd|resisc|eurosat|pcam|imagenet-s|imagenet-o|average|
> |-----|-----|---------|---------|---------|---------|---------|---------|---------|---------|---------|---------|---------|-----|
> |AM-RADIO|75.38|95.04|71.48|78.47|25.66|83.62|45.37|38.48|31.35|50.02|51.34|69.45|58.21|
> |ours           |75.52|96.27|77.04|84.32|50.31|93.40|55.74|63.83|63.83|52.68|59.67|34.90|66.54|
>
>
> ___
> **Reference**
>
> - Ranzinger et al. "Am-radio: Agglomerative vision foundation model reduce all domains into one." CVPR (2024).

---

### Official Review · Reviewer_ue7L · 2025-03-16

**Overall Recommendation:** 3

**Summary:**

This paper employs a kernel-based method to align CLIP's visual representation with DINOv2's representation. The goal is to enhance the CLIP vision encoder's fine-grained perception capabilities. The resulting aligned visual encoder demonstrates improvements in zero-shot object recognition, fine-grained spatial reasoning, and localization abilities.

## Update after rebuttal
My major concern is addressed during the rebuttal. Thanks the authors for the additional experiments.

**Claims And Evidence:**

While the evidence presented is clear, it is not fully convincing in supporting the paper's claims.

Claim 1: The aligned encoder shows improvements on vision-centric tasks.

Evidence 1: The paper evaluates performance on zero-shot object recognition, image-to-text and text-to-image retrievals, counting, spatial reasoning, and caption recognition and localization ability. However, the tables primarily compare results to the original CLIP encoder. The claim would be more convincing if comparisons with the DINO v2 encoder were included for all these vision-centric tasks.

Claim 2: The aligned encoder shows improvements on MLLMs.

Evidence 2: The paper evaluates performance on LLaVA and OpenFlamingo. However, for Table 5, the distinction between baseline and PEFT is unclear. Since the baseline LLaVA-1.5-7B already undergoes two-stage training (pretraining and SFT), the rationale for conducting additional PEFT using the same tuning dataset needs explanation.

**Essential References Not Discussed:**

All are discussed.

**Experimental Designs Or Analyses:**

The experimental design is validity. The ablation study is solid.

**Methods And Evaluation Criteria:**

Yes, the kernel-based method makes sense. What if we removed the regularization term? How would that affect the MLLM performance?

**Other Comments Or Suggestions:**

No.

**Other Strengths And Weaknesses:**

Strength: use kernel methods to combine the strength of CLIP and DINOv2.

Weakness: no weakness.

**Questions For Authors:**

1. Can you compare the performance of MLLM using CLIP encoder, DINOv2 encoder and your aligned encoder following the standard 2 stage method?

2. Can you please explain more on “projection” and “feature” methods in table 1?

3. This paper use the kernel method to align CLIP to DINOv2. What about aligning the DINO v2 with CLIP model? Using DINO v2 as baseline, use kernel method to align it with the CLIP to get the encoder.

**Relation To Broader Scientific Literature:**

As mentioned in the paper, kernel methods have also been applied to evaluating the fidelity, diversity, and novelty of generative models.

**Theoretical Claims:**

The two proofs are correct.

---

> ### Author Rebuttal · Authors · 2025-03-31
>
> We thank Reviewer ue7L for the thoughtful and constructive feedback on our work. The following is our response to the comments and questions in the review.
> ___
> > **Comparison with DINOv2 encoder**
>
> We would like to clarify that our numerical evaluation of the fine-tuned CLIP using alignment with DINOv2 performs experiments of the following two types of image-centric models:
> 1. Text-image cross-modality tasks such as zero-shot classification and image-text retrieval: We note that DINOv2 is not applicable to these tasks, since they require a text encoder that is not existing in DINOv2. This is why we did not report the performance of DINOv2.
> 2. Image-only tasks: In these experiments, our goal is to highlight that aligning CLIP with image-modality expert DINOv2 enhances CLIP's visual performance. Per the reviewer's suggestion, we include DINOv2's performance on linear probing and probing bench. Results show DINOv2 significantly outperforms CLIP on purely perceptual tasks (e.g., spatial reasoning, locality probing), with alignment successfully narrowing this gap. For tasks requiring both visual and multimodal abilities (e.g., caption recognition), the aligned model leverages strengths from both models, outperforming each individual model.
>
> In both settings, we aim to show that aligning with modality experts improves multi-modal models, that is why we aligned CLIP with DINOv2 rather than the opposite, and focused on the comparison with the original CLIP. We will include the additional comparison with DINOv2 in the revised text.
> |Task|svhn|gtsrb|clevr distance|clevr counts|probing local|probing global|
> |-|-|-|-|-|-|-|
> |CLIP|65.20|72.49|22.97|41.25|46.40|54.51|
> |DINOv2|49.00|61.25|47.81|65.43|49.75|57.01|
> |ours|69.39|74.51|30.82|49.67|47.44|55.33|
> ___
> > **The explanation for additional PEFT**
>
> Thank you for pointing this out. Since we have fine-tuned the visual encoder, we also attempted to finetune the downstream component so that we can achieve even a stronger improvement. Because our alignment method ensures the visual encoder does not deviate significantly from the original model, fine-tuning the downstream component can be efficiently performed using PEFT. Following the reviewer’s suggestion, we further experimented with standard two-stage training, and compared the CLIP encoder w/wo alignment. In our experiments, we observed that this led to even greater performance and confirmed that the alignment improves the MLLMs performance.
> |Task|AI2D|text-ocr|text-pure|pope|tallyQA|vsr|RefCOCOg|RefCOCO+|RefCOCO|vqa-v2|llava-wild|llava-coco|average|
> |-|-|-|-|-|-|-|-|-|-|-|-|-|-|
> |w/o align|53.27|57.82|48.63|85.45|61.45|51.47|55.19|51.97|60.49|76.40|42.97|75.60|60.06|
> |w/ align|53.95|58.89|50.51|86.05|62.14|52.55|58.68|57.74|65.79|76.97|52.20|78.90|62.86|
>
> Due to the time limit, we could not finish the experiment with DINOv2. However, [[Karamcheti et al., 2024](https://arxiv.org/abs/2402.07865)] has shown that DINOv2’s MLLM performance does not match that of CLIP, which may due to its image-only pre-training protocol.
> ___
> > **Explanations  on “projection” and “feature” methods**
>
> The term “projection” in Table 1 refers to the case with a trained projection head to project the DINOv2 features into the CLIP feature space. Specifically, we freeze the CLIP and DINOv2 encoders, and train a projection head $P(\cdot)$, to minimize the ${\mathbb{E}}[\Vert P(g(I_i)) - f(I_i) \Vert^2]$. Then we use $P(g(I_i))$ as the visual representation to conduct zero-shot classification. The term “feature” refers to aligning the CLIP and DINOv2 representations directly, subject to a linear transformation. Formally, we fine-tune the CLIP encoder to minimize the following objective:
>
> $\min_{\theta, \mathbf{R}} {\mathbb{E}}[w\Vert f_\theta(I_i) - \mathbf{R}g(I_i)\Vert_2^2 + \Vert f_\theta(I_i) - f_{\theta_0}(I_i)\Vert_2^2]$
>
> We will make it more clear in the revised text.
> ___
> > **Effects of the regularization term**
>
> Regarding the reviewer's suggestion, we would like to note that there is an ablation study of the regularization term in Fig. 3(b). Removing the regularization term results in a significant drop of the zero-shot classification performance, which shows its importance for maintaining the visual-text alignment. For a more detailed evaluation, we have run a new experiment where we plugged the encoder fine-tuned without regularization into LLaVA without fine-tuning the LLM. The results below also indicate the performance drop. We will include the additional results in the revised paper.
> |Task|AI2D|text-ocr|text-pure|pope|tallyQA|vsr|RefCOCOg|RefCOCO+|RefCOCO|vqa-v2|llava-wild|llava-coco|average|
> |-|-|-|-|-|-|-|-|-|-|-|-|-|-|
> |w/o reg.|52.10|50.22|29.90|83.47|57.73|51.47|13.72|11.23|12.38|71.32|38.57|70.47|45.22|
> |w/ reg.|53.66|55.70|41.53|84.92|60.39|51.50|14.15|12.89|15.37|75.10|44.67|76.60|48.87|
> ___
> **Reference**
> - Karamcheti et al. "Prismatic vlms: Investigating the design space of visually-conditioned language models." ICML (2024).

---

> > ### Comment · Reviewer_ue7L · 2025-04-04
> >
> > Hi Author,
> >
> > Thank you for your efforts and additional experiments. I'd like to reply to the rebuttal and discuss more with the author.
> >
> > **Comparison with DINOv2 encoder.** Thanks for you experiment. This address my concern. I have another question about this part. This method involves training a new vision encoder that using kernel methods to align CLIP and Dinov2. There are methods, like Additive-MoF(Eyes Wide Shut, Cambrain-1), that directly concatenate the features of CLIP and DINOv2 also shows good performance. The benefit of their method is no need to train a new vision encoder. Would the performance of this method beats their concatenate method?
> >
> > **The explanation for additional PEFT.** Thanks for your experiment. This question is addressed.
> >
> > **Explanations on projection and feature methods.** Thanks for your explaination. This issue is addressed.
> >
> > **Effects of the regularization term.** The regularization term seems important. This issue is addressed.

---

> > > ### Author Response · Authors · 2025-04-08
> > >
> > > We thank Reviewer ue7L for the thoughtful feedback on our responses and are glad to hear that they could address most of the reviewer’s comments.
> > >
> > > Regarding the follow-up question, we would like to first clarify that a concatenation of CLIP and DINOv2 image encoders is not generally an alternative to our proposed method of kernel-based aligning CLIP to DINOv2. Although the concatenation of the image encoders can potentially improve the visual capability, the DINOv2 model lacks a text encoder, leading to a loss in the alignment between CLIP text embedding and the concatenated image embeddings. Therefore, the concatenated embedding can no longer apply to cross-modal applications, such as zero-shot classification and image-text retrieval. On the other hand, our kernel-based fine-tuning method is designed to mitigate the loss in the alignment of the image and text encoders, resulting in the improvement of the visual performance of CLIP while not sacrificing the alignment with the original text encoder.
> > >
> > > Also, regarding the MLLMs application in the references mentioned by the reviewer, we have performed additional comparisons of our aligned CLIP  with the Additive-MoF method, and the results are summarized in the following table.
> > >
> > > |MLLMs Task|AI2D|text-ocr|text-pure|pope|tallyQA|vsr|RefCOCOg|RefCOCO+|RefCOCO|vqa-v2|llava-wild|llava-coco|average|
> > > |-|-|-|-|-|-|-|-|-|-|-|-|-|-|
> > > |Additive-MoF|51.36|45.93|18.66|86.58|63.18|51.47|68.44|65.34|69.69|75.12|41.54|74.50|59.32|
> > > |ours|53.95|58.89|50.51|86.05|62.14|52.55|58.68|57.74|65.79|76.97|52.20|78.90|62.86|
> > >
> > > While Additive-MoF obtained better performance in some of these MLLMs tasks, e.g. object localization, it performed suboptimally for the other tasks, e.g. open-ended visual question answering. As mentioned in the original paper by [[Tong et al., 2024](https://arxiv.org/abs/2401.06209)], a straightforward combination of CLIP and DINOv2 features faces an inherent trade-off between visual grounding accuracy and instruction-following capability. On the other hand, our alignment fine-tuning approach achieves enhanced visual capabilities without compromising alignment to the textual embedding space. Furthermore, we note that our framework does not necessitate complete re-tuning of the LLM component, making it particularly advantageous in resource-constrained scenarios.
> > >
> > > We will include the above discussion and results in the revised text to better clarify our motivation behind the proposed kernel-based alignment approach.
> > > ___
> > > **Reference**
> > > - Tong et al. "Eyes wide shut? exploring the visual shortcomings of multimodal llms." CVPR (2024).

---

### Decision · Program_Chairs · 2025-05-01

**Decision:**

Accept (poster)

**Comment:**

The reviewers recognise the authors'  method to better align VLM image-representations with those of vision only models in order to  efficiently improve sensitivity to image features, particularly fine grained features, not fully captured by the original text alignment objective. This is exemplified and evaluated using CLIP's image representations with DINOv2's. Recognised strengths of the paper include:
* The paper is clearly written and easy to follow. The illustrations and tables are also clear.
* The alignment objective is well-motivated and can resolve the feature conflict problem when aligning two different forms of features.
* The experimental design is appropriate and the ablation study is solid.
* The experimental results are comprehensive and support the claims.

Identified limitations and concerns include:
* Some explanations could have been made clearer to reviewers not familiar with kernel methods and representational alignment literature.
* The improvements are consistent across conditions, but at times small in size.

Note that additional results presented in discussion by the authors show additional improvements across a range of conditions which will strengthen the paper by their inclusion. Moreover, the theoretical explanations, justifications and results appear sound and the approach novel. However, no one reviewer championed the paper all only recommending "weak accept", and I have reflected this in my recommendation.